# Effect of Micro-Nanobubbles on Arsenic Removal by *Trichoderma atroviride* for Bioscorodite Generation

**DOI:** 10.3390/jof9080857

**Published:** 2023-08-17

**Authors:** Asunción Guadalupe Morales-Mendoza, Ana Karen Ivanna Flores-Trujillo, Jesús Adriana Ramírez-Castillo, Salvador Gallardo-Hernández, Refugio Rodríguez-Vázquez

**Affiliations:** 1Doctoral Program in Nanosciences and Nanotechnology, Center for Research and Advanced Studies of the National Polytechnic Institute (CINVESTAV-IPN), Instituto Politécnico Nacional Avenue, No. 2508, Zacatenco, Mexico City 07360, Mexico; asuncion.morales@cinvestav.mx; 2Department of Biotechnology and Bioengineering, Center for Research and Advanced Studies of the National Polytechnic Institute (CINVESTAV-IPN), Instituto Politécnico Nacional Avenue, No. 2508, Zacatenco, Mexico City 07360, Mexico; ivanna.flores@cinvestav.mx (A.K.I.F.-T.); adriana.ramirez@cinvestav.mx (J.A.R.-C.); 3Subdirection of Health Riks, National Center of Disasters Prevention CENAPRED, Delfin Madrigal Avenue, No. 665, Pedregal de Santo Domingo, Coyoacán, Mexico City 04360, Mexico; 4Departament of Physics, Center for Research and Advanced Studies of the National Polytechnic Institute (CINVESTAV-IPN), Instituto Politécnico Nacional Avenue, No. 2508, Zacatenco, Mexico City 07360, Mexico; salvador.gallardo@cinvestav.mx

**Keywords:** arsenic, ferrihydrite, micro-nanobubbles, mycoremediation, bioscorodite, *Trichoderma atroviride*, Fe, ORP

## Abstract

The global environmental issue of arsenic (As) contamination in drinking water is a significant problem that requires attention. Therefore, the aim of this research was to address the application of a sustainable methodology for arsenic removal through mycoremediation aerated with micro-nanobubbles (MNBs), leading to bioscorodite (FeAsO_4_·2H_2_O) generation. To achieve this, the fungus *Trichoderma atroviride* was cultivated in a medium amended with 1 g/L of As(III) and 8.5 g/L of Fe(II) salts at 28 °C for 5 days in a tubular reactor equipped with an air MNBs diffuser (TR-MNBs). A control was performed using shaking flasks (SF) at 120 rpm. A reaction was conducted at 92 °C for 32 h for bioscorodite synthesis, followed by further characterization of crystals through Fourier–Transform Infrared Spectroscopy (FTIR), Scanning Electron Microscopy (SEM), and X-ray diffraction (XRD) analyses. At the end of the fungal growth in the TR-MNBs, the pH decreased to 2.7–3.0, and the oxidation-reduction potential (ORP) reached a value of 306 mV at 5 days. Arsenic decreased by 70%, attributed to possible adsorption through rapid complexation of oxidized As(V) with the exchangeable ferrihydrite ((Fe(III))_4-5_(OH,O)_12_), sites, and the fungal biomass. This mineral might be produced under oxidizing and acidic conditions, with a high iron concentration (As:Fe molar ratio = 0.14). The crystals produced in the reaction using the TR-MNBs culture broth and characterized by SEM, XRD, and FTIR revealed the morphology, pattern, and As-O-Fe vibration bands typical of bioscorodite and römerite (Fe(II)(Fe(III))_2_(SO_4_)_4_·14H_2_O). Arsenic reduction in SF was 30%, with slight characteristics of bioscorodite. Consequently, further research should include integrating the TR-MNBs system into a pilot plant for arsenic removal from contaminated water.

## 1. Introduction

Water pollution by heavy metals is a globally significant issue. Arsenic (As), a metalloid, is considered an extremely toxic and persistent contaminant that accumulates rapidly in organisms [1]. As can accumulate within the food chain, leading to environmental damage and health issues [2,3]. Its high toxicity and carcinogenicity contribute to various problems concerning food safety in different countries [4].

The presence of As occurs mainly through two pathways. Firstly, this element can naturally be found in groundwater at high concentrations or as part of rocks, and it is also released through volcanic activity. Secondly, anthropogenic sources contribute to As contamination, including industrial, mining, and agricultural activities, such as the use of pesticides and the addition of arsenic as a feed additive for livestock and poultry [3,5,6,7,8].

Arsenite (As(III)) and arsenate (As(V)) salts are inorganic As ions that predominate depending on the type of oxidation-reduction potential (ORP) of the soil. During rainy seasons and floods, As(III) exhibits greater mobility and toxicity compared to As(V), especially under acidic pH conditions [4,9,10]. Furthermore, these two inorganic forms are also present in water, where As(III) can be found as the dominant species under reducing conditions, while As(V) is identified as a more stable species in oxygenated environments [9].

Due to the potential repercussions of As, strategies for its removal have been investigated, including ion exchange, adsorption, and activated carbon, among others. These procedures have shown high effectiveness in reducing As concentrations. However, they are associated with high costs [10]. However, one of the most emerging and promising technologies is the production of scorodite crystals (FeAsO_4_·H_2_O) for the immobilization of this metalloid [11]. It is also important to mention that As(V) is easier to immobilize than As(III), which is why it is necessary to perform oxidation using strong oxidants [10].

Scorodite is a secondary iron mineral that contains arsenic bound to its surface, allowing for the storage and immobilization of this contaminant [4]. It is mainly found in impacted areas or As deposits, resulting from the oxidizing microbial activity of Fe(III) [10]. Scorodite produced by microorganisms (bioscorodite), also known as secondary scorodite, contains crystals that closely resemble the mineral scorodite. It can play an essential role in the environment for controlling high concentrations of As detected in contaminated sites. Moreover, bioscorodite is characterized by its stability, which makes it a suitable strategy for the remediation of sites contaminated with As [11]. The formation of bioscorodite is obtained at temperatures of 80 °C, low pH, and in solutions with high concentrations of arsenate and iron [12].

Among the different biological mechanisms for the formation of bioscorodite, the studies carried out by González-Contreras et al. (2010, 2012) [11,12] are included, in which extremophilic bacteria were used to convert As(V) and Fe(II) into bioscorodite. The detoxification of As in a biological system, as well as other contaminants, is possible due to the biodegradation, biotransformation, or biomineralization processes carried out by various microorganisms, particularly fungi [13,14]. Through biochemical methylation reactions with As, fungi can be used to achieve the detoxification of arsenic [2,5]. Recently, the detoxification of arsenic has been reported through the formation of solid particles by the mycorrhizal fungus *Aspergillus flavus* [2]. Several fungi have also been reported to possess high tolerance and arsenic removal capacity, including *Aspergillus candidus* [15], *Fusarium oxysporum* [16], *Talaromyces* sp. [5], *Trichoderma* sp., *Neocosmospora* sp., and *Rizhopus* sp. [17]. Due to their high potential as a remediation method, as they are environmentally friendly and cost-effective, fungi have gained attention. One of the most used fungal genera for bioremediation is *Trichoderma* spp., which is commonly found in soil and plant roots. This genus of fungi is utilized due to its broad tolerance to various contaminants, including heavy metals, pesticides, aromatic hydrocarbons, and organometallic compounds, among others [18].

Different chemical and biological strategies have been investigated for the proper disposal of arsenic. Recently, Ramírez-Castillo et al. (2023) [19] produced the biomineral bioscorodite through the reaction of As(III) and Fe(II), carried out at 92 °C in the *T. atroviride* culture medium. This culture medium was selected to lower the pH to 2.0, favoring the activity of the enzyme glucose oxidase, and consequently, the production of H_2_O_2_, which was correlated with an increase in the ORP up to 306 mV, thereby promoting bioscorodite production. The conditions of these oxidation processes could be enhanced by introducing oxidizing agents through aeration with micro-nanobubbles (MNBs). These bubbles have gained significance in various fields, including water treatment, due to the diverse properties of microbubbles (ranging in size from 1–100 µm) and nanobubbles (sizes smaller than 1 µm) [20,21]. These properties include high stability, longevity, rapid adherence to hydrophobic solid surfaces, oxygen transport, and generation of reactive oxygen species [22]. Micro-nanobubbles have potential applications in various areas such as medicine, agriculture, and the acceleration of metabolism in different species, among others [23]. There are different methods for producing MNBs [24] and application processes using specific equipment to enhance oxygen transfer, such as Airlift reactors in conjunction with microorganisms for the bioremediation of sites contaminated with various toxic compounds [25].

Based on the above-mentioned information, the hypothesis of this study was that aeration with MNBs improves the oxidation-reduction potential of the *T. atroviride* culture medium, enhancing the oxidation of As(III) and Fe(II) for bioscorodite production, compared to mechanical shaking. Thus, the main objective of this research was to investigate the impact of MNBs aeration on the cultivation and mycoremediation potential of *T. atroviride* cultures for bioscorodite generation.

## 2. Materials and Methods

### 2.1. Microorganism and Growth Conditions

The strain of *Trichoderma atroviride* CDBB-H-1125 was provided by the Strain Collection of the Center for Research and Advanced Studies of the National Polytechnic Institute (CINVESTAV). The inoculum production was carried out in modified Wunder medium [26,27]: glucose (10 g/L), polypeptone (1 g/L), (NH_4_)_2_SO_4_ (1 g/L), MgSO_4_·7H_2_O (0.5 g/L), KH_2_PO_4_ (0.875 g/L), K_2_HPO_4_ (0.125 g/L), CaCl_2_·2H_2_O (0.1 g/L), NaCl (0.1 g/L), MnSO_4_·H_2_O (0.02 g/L), and FeSO_4_·7H_2_O (0.001 g/L). A total of 170 mL of medium were added to each Erlenmeyer flask, and 6 discs of 6 mm in diameter of the fungus previously grown on Potato Dextrose Agar (PDA) medium were added. The inoculated flasks were maintained under agitation at 125 rpm, 28 °C, for 4 days. After the incubation period, the biomass was filtered and used in further experiments.

### 2.2. Set Up of the Micro-Nanobubbles Aeration

Figure 1 presents the tubular reactor utilized for MNBs generation, comprising borosilicate columns with a capacity of 450 mL. The reactor incorporates a porous diffuser for MNBs production, a process initiated as the air flow traverses the porous diffuser. In order to determine the bubble size, the following procedure was performed: the average bubble diameter size was evaluated by adding 100 mL of milli-Q water. Subsequently, the aeration system from the main line was connected and maintained for 60 min, and samples were analyzed using Zetasizer Nano Malvern equipment, performing measurements from 0 to 10,000 nm. The refractive index for water was considered 1.33 at 25 °C, the sample was read in triplicate, and the average value was taken as the final value.

### 2.3. Determination of Culture Parameters of the Fungus T. atroviride

The parameters of the fungal culture were evaluated using two systems: SF and TR-MNBs with modified Wunder medium, 1 g/L sodium arsenite (NaAsO_2_) (J.T. Baker 99%), and 8.5 g/L ferrous sulfate (FeSO_4_·nH_2_O) (Reasol, 99%). Particularly, in the SF system, 0.15 g (dry weight) of fungal biomass was added to each Erlenmeyer flask, which already contained 100 mL of Wunder medium. The flasks were then placed in a shaker incubator, set at 125 rpm and 28 °C, and left to incubate for 5 days. Samples were collected daily for subsequent analyses. On the other hand, in the TR-MNBs system, 0.3 g (dry weight) of fungal biomass was added to bubbling columns containing 200 mL of the same medium as mentioned above. The redox potential (mV), fungal growth (mg/L), glucose oxidase enzyme (U/mL), and pH were analyzed daily in the fungal culture broth for a period of 5 days in both systems.

### 2.4. Biochemical Parameters

#### 2.4.1. Residual Glucose

Residual glucose was determined as the reducing sugar using the 3,5-dinitrosalicylic acid (DNS) method [28]. A total of 1 mL of the previously filtered medium from the TR-MNBs and SF culture system was used, 1 mL of DNS reagent was added, and the mixture was left to react for 5 min at 80 °C. Subsequently, 4 mL of distilled water was added. The absorbance was measured at 575 nm in a Shimadzu UV-VIS UV-1800 spectrophotometer.

#### 2.4.2. Glucose Oxidase Enzyme Activity

Glucose oxidase enzyme activity was evaluated by adding 50 µL of 1 mg/mL glucose, 950 µL of culture medium extract, and 1 mL of 0.2 M ABTS-Phosphate buffer, pH = 5.5, with an ABTS concentration of 0.41 mM and 1 mL of 5 U/mL POD in 40 mM phosphate buffer at pH 6.8. The mixture was incubated for 10 min at 37 °C, and the absorbance was measured at 410 nm [29].

#### 2.4.3. Biomass

Biomass quantification was performed by vacuum filtering through a weighed Whatman No. 40 filter paper. The biomass retained on the paper was dried in an oven at 60 °C for 1 h, and the filters containing the biomass were weighed again to determine the dry weight basis of the biomass. The filtered culture medium was used for further analysis.

### 2.5. Physicochemical Parameters

The hydrogen potential (pH), dissolved oxygen concentration (DO in mg/L), and ORP (in mV) were determined using the HANNA multiparameter equipment (HI9829).

### 2.6. Scorodite Production

#### 2.6.1. Chemical Scorodite

Chemical scorodite seed was obtained in 250 mL in a balloon flask by adding 0.24 g Fe(II) (FeSO_4_·nH_2_O) to 50 mL of distilled H_2_O. The pH was adjusted to 1.3 using concentrated H_2_SO_4_, followed by the addition of 0.28 g of As(V) (Na_2_HAsO_4_·7H_2_O), resulting in a milky, yellow-colored solution. Subsequently, 0.56 g of gypsum (CaSO_4_·2H_2_O) (J.T. Baker, 99%) was introduced as a seed. Finally, 10 drops of concentrated H_2_SO_4_ were added to lower the pH to 1.3, as acidic conditions are necessary for scorodite synthesis.

The flask was positioned on a stirring rack, equipped with a condenser to prevent vapor condensation and minimize volume loss. The temperature was maintained at 92 °C for a duration of 21 h. Upon completion of the reaction, the solution was allowed to cool, filtered through Whatman No. 40 filter paper, and subsequently placed in a desiccator at room temperature.

#### 2.6.2. Bioscorodite

Bioscorodite production was conducted within the SF and TR-MNBs systems. For this purpose, 100 mL of *T. atroviride* culture medium, obtained from 5-day cultivation for subsequent experiments, was added to a balloon flask. The pH was adjusted to 1.3 using concentrated H_2_SO_4_. Subsequently, 0.2 g of scorodite seed (FeAsO_3_·2H_2_O) was introduced. The flask was then placed in a cooler to prevent evaporation loss and maintained at 120 rpm at 92 °C for a duration of 32 h. Finally, the solution was allowed to cool and was filtered through a Whatman No. 40 filter paper.

### 2.7. Characterization of Chemical Scorodite and Bioscorodite

#### 2.7.1. Fourier Transformation Infrared Spectrophotometry (FTIR)

The chemical scorodite and bioscorodite reaction samples were analyzed using Fourier Transform Infrared Spectroscopy on a Nicolet 6700 instrument to determine the functional groups present in chemical scorodite and bioscorodite. Spectral scanning was performed in the 4000–400 cm^−1^ range with a resolution of 2 cm^−1^. The data were analyzed from the signals obtained.

#### 2.7.2. Scanning Electronic Microscopy (SEM)

Morphological characterization of the filtered solids from the chemical scorodite and bioscorodite synthesis was performed. The analysis was conducted using scanning electron microscopy (SEM) with an Oxford Instrument. The samples were prepared by adding a light layer of gold and analyzed under a high vacuum at 30 kV, with a magnification of ×5000.

#### 2.7.3. X-ray Diffraction Analysis

X-ray diffraction spectra of chemical scorodite and bioscorodite were obtained using Rigaku-Smart Lab equipment with a copper objective at 40 kV and 100 mA. The scan was performed from 10 to 100° in increments of 0.02°.

### 2.8. As Determination

The remaining As concentrations after the oxidation reaction were quantified using Atomic Absorption Spectroscopy (Shimadzu 6300, Shimadzu Scientific Instruments, Columbia, MD, USA). For each measurement, the average of the three replicates was considered. An As standard (J.T. Baker) was used for the calibration graph in the atomic absorption equipment of the National Center for Disaster Prevention.

### 2.9. Statistical Analysis

A multiple comparison of means was carried out by Least Square Differences (LSD) with a significance level (α) of 0.05, using the Statgraphics Centurión XVI.I statistical software for the physicochemical analyses during the culture of *T. atroviride*.

## 3. Results

### 3.1. Nanobubbling Size Determination in the TR-MNBs System

Using Dynamic Light Scattering, the particle size distribution of the bubbles in the water samples were determined in the TR-MNBs system. Figure 2 displays the peaks observed in each reading (shown in blue, red, and green), mainly attributed to the dynamics of the bubbles generated through Brownian movement, which allows them to move randomly, collapse, or group together. As a result, it is inferred that bubbles of different sizes can be found in the system, with the most predominant bubble size recorded as an average of each reading at 625.8 nm, followed by sizes of 3.5 nm and 127.1 nm (Table 1). These three prominent sizes are highlighted, showing higher intensities compared to other signals in the bubbled water.

The analysis indicates that the system generates bubbles within both the micrometer and nanometer scales. According to the Fine Bubbles Industries Association (FBIA), microbubbles have diameters ranging from 1 to 100 μm, while nanobubbles are defined as those with diameters smaller than 1 μm. When air is delivered through a fine-pored membrane, both types of bubbles, micro-bubbles and nanobubbles, are present, and hence, they are referred to as micro-nanobubbles (MNBs). The presence of these different bubble sizes indicates that microbubbles have a higher likelihood of rupturing, leading to the generation and/or collapse of smaller bubbles (nanobubbles). However, there is also a possibility of coalescence, where nanobubbles combine and form larger bubbles, such as MNBs [30].

### 3.2. Glucose Consumption, Glucose Oxidase Activity and Biomass Produced

Glucose consumption in both the TR-MNBs and SF systems exhibited a decrease in glucose concentration (Figure 3a). No significant differences between the two systems were observed during the culture, except at the 5-day mark, where glucose in the SF decreased to 4 g/L h, representing a 61% decrease compared to the initial concentration. In comparison, the TR-MNBs system showed a reduction to 2.6 g/L, indicating a 75.50% decrease of glucose consumption. Similarly, the glucose oxidase activity did not show significant differences between the two systems and increased over time (Figure 3b). Glucose oxidase activity displayed a negative correlation with glucose concentration (Figure 3b), indicating that as glucose concentration decreased in the medium, glucose oxidase activity increased (R^2^ = −0.96 for TR-MNBs and −0.86 for SF). On the other hand, glucose consumption exhibited a positive correlation with the decrease in pH (R^2^ = 0.95 for TR-MNBs, and 0.89 for SF). A previous report [31] stated that glucose oxidase is an enzyme classified as an FAD-dependent glycoprotein, catalyzing the oxidation of D-glucose, yielding gluconic acid and hydrogen peroxide as primary metabolites. Glucose oxidase activity increased over time in both systems (Figure 3b). This phenomenon occurs because as the culture time lengthens, the enzyme becomes more active in oxidizing the glucose molecule in the medium, as evidenced by its positive correlation with the ORP (R^2^ = 0.98 for TR-MNBs and 0.96 for SF). Figure 3c depicts the biomass produced during the 5-day culture, revealing significant differences between the two systems. TR-MNBs exhibited the highest biomass production (5.3 g/L). Aeration with MNBs promotes fungal metabolism, consequently enhancing biomass production. This correlation (*p* < 0.05) aligns with the increase in glucose consumption and the remaining glucose concentration at the conclusion of the fungal culture. Hence, an appropriate glucose concentration can positively impact fungal cell metabolism; however, excessive concentration can lead to metabolite overproduction. The negative correlations between biomass and residual glucose were observed (R^2^ = 0.84 for TR-MNBs and −0.85 for SF), signifying that greater biomass necessitates a higher glucose concentration, a key carbon source employed by filamentous fungi [32].

### 3.3. Determination of Physicochemical Parameters

The physicochemical parameters of the fungus culture medium are shown in Figure 4. A decrease in pH is observed (Figure 4a), and this parameter determines the degree of alkalinity or acidity in a system and also directly affects the development of certain microorganisms [33]. This behavior is considered desirable because it is a necessary condition for the synthesis of scorodite, as reported by González et al. (2010) [12]. Figure 4a shows the pH results for the two systems over 5 days.

A decrease in the pH was observed from the first day of the experiment compared to the initial time. In the case of SF, the pH dropped by two units from the first day. Similarly, the pH values (2.7 and 3.0) did not show significant differences between both systems, and by day 5, they reached the lowest value, which is favorable for producing bioscorodite. In other conditions, the production of bioscorodite requires a pH around 2 and a temperature close to 70 °C for its formation [34]. The decrease in pH is attributed to the ability of *Trichoderma* to produce organic acids through the degradation or hydrolysis of glucose [13,35,36]. The behavior of DO during cultivation is illustrated in Figure 4b. DO serves as an indicator of the quantity of gaseous oxygen within a liquid medium and is of paramount importance in water quality and treatment [37]. It plays a vital role in aerobic respiration and the regulation of the ORP [38]. Figure 4c presents the ORP behavior, where not-significant differences were observed between the two cultures. At the beginning of the experiment, an abrupt decrease in the DO in the medium was observed in each of the systems. In the following days, the DO showed low fluctuations, then increased again at the 5-day mark.

Likewise, one of the most important parameters in the development of the oxidation reaction is mainly due to the presence of As(III) and Fe(II) ions in the media. Figure 4c illustrates the behavior of this parameter during the experiment for both systems. It was observed that this parameter increased at a higher rate in the TR-MNB than in the SF. However, after 2 days, an increase in the ORP was observed, reaching 306 mV in the TR-MNBs system, while in SF, the ORP was 266 mV. Throughout the experiment, the ORP measurements in the SF were found to be in the range of 270–250 mV, whereas in the TR-MNBs system, there were variations, with a maximum ORP of 306 mV.

### 3.4. Bioscorodite Characterization

#### 3.4.1. Fourier Transform Infrared Spectroscopy

Figure 5 displays the FTIR spectra along with the corresponding SEM analyses of bioscorodite and chemical scorodite. Bands with heightened absorbance were observed within the wavelength range of 3000–3500 cm^−1^, which correspond to the stretching bond of –OH [11,39] (Figure 5a–c). Similarly, peaks within the 1750–1500 cm^−1^ range can be attributed to the vibrational bonding of H_2_O molecules [40]. More specifically, the signals at 562 and 465 cm^−1^ can be attributed to the Fe-O bond [41]. The signal at 838 cm^−1^ corresponds to the stretching bonds of As–O–Fe [42], although this signal was relatively faint in crystals from the TR-MNBs system (Figure 5b). Additional peaks in the vicinity of 422–434 are distinctive to vibrations of As-O binding, a characteristic feature of scorodite [12,42]. Furthermore, it is pertinent to mention that all three spectra exhibit frequencies between 400–1300 cm^−1^, corresponding to active vibrations bonds of SO_4_, AsO_4_, and H_2_O/OH [43]. Notably, prominent vibrations at 516, 510, and 509 cm^−1^ correspond to Fe-O-As ligand vibrations [43].

#### 3.4.2. Scanning Electron Microscopy (SEM)

The bioscorodite SF (Figure 5a’) exhibits crystals of different sizes, with apparent rectangular shapes distributed among them, and small round crystals are also observed. In contrast, the chemical scorodite (Figure 5c’) displays crystals in rectangular clusters with round edges and larger crystal sizes. On the other hand, bioscorodite obtained from TR-MNBs aeration is presented in clusters with spherical shapes and nanometer sizes (Figure 5b’). Similar irregular formations and clusters have been reported in previous studies [44].

#### 3.4.3. X-ray Diffraction

Figure 6 shows the diffractogram patterns of scorodite, which are consistent with the bioscorodite obtained from other sources [45,46,47,48]. The highest intensity peaks in the range of 15–21 (2θ°), particularly at 23.4, 28.1, 29.2, 29.7, 34.5, and 34.8 (2θ°), match those reported by González-Contreras et al. (2010) [12]. These signals were found in both bioscorodite and chemical scorodite diffractograms (Figure 6). The highest intensity peaks were observed in the bioscorodite of the SF system (Figure 6a) and chemical scorodite (Figure 6c). Similarly, the sharpness and definition of the peaks are related to the presence of larger crystals and greater crystallinity. Moreover, it is worth noting that different authors [49,50] have reported the synthesis of scorodite with patterns similar to those found in this work, where the signals are present in the 25–40 (2θ (°)) range, specifically at 26, 27, 29, 30, 34, and 38 (2θ (°)), indicating a greater number of peaks but with lower intensity compared to the previously mentioned range. The alignment of the peaks for scorodite in this investigation confirms its presence. However, those peaks were of greater intensity in the bioscorodite diffractogram of the TR-MNBs system (Figure 6b). The formation of scorodite is consistent with previous observations.

### 3.5. Determination of As in the Culture Medium of T. atroviride

The arsenic removal in the TR-MNBs was 12.5% at day 2, while in the SF only 2% was achieved. However, at day 5, a significant improvement (70%) was obtained in the TR-MNBs, compared to 30% As removal in the SF. Both As removal values at day 2 suggest that this metalloid was adsorbed into the *T. atroviride* biomass. Furthermore, a final measurement of the total As was made after the formation of scorodite, where an apparent increase in As concentration was observed. This increase was attributed to the As coming from the chemical scorodite seed added to the liquid medium prior to the precipitation reaction. Therefore, it is recommended to use other seeds, such as calcium sulfate, to avoid this problem.

Samples taken at the initial time, 2, and 5 days from SF culture media, where the difference between the initial samples was noticeable, exhibited higher solids formation with more suspended solids remaining than in the TR-MNBs medium, which did not present sediments.

## 4. Discussion

The monitoring of glucose consumption, glucose enzyme activity, biomass production, and physicochemical parameters in the two evaluated systems offered comprehensive insights into the impact of MNBs on the behavior of *T. atroviride* culture amended with As(III) and Fe(II) salts in the Wunder medium.

In *Trichoderma* species, it has been reported that they have adequate nutrient transport systems and different enzyme complexes that allow them to obtain the carbon source from the environment to survive [51]. In this study, an abrupt decrease in glucose concentration was observed during the initial days of cultivation, suggesting that *T. atroviride* could utilize glucose as a carbon source, since this sugar serves as a primary substrate [52].

In contrast, *T. atroviride* causes changes in physicochemical parameters, particularly in the pH. The decrease in this parameter is mainly caused by the synthesis of organic acids from the degradation or hydrolysis of glucose [35,36]. Fungal species are known to produce various organic acids, including citric, oxalic, gluconic, and malic [13]. In the context of the experiments conducted, the decrease in pH could be attributed to the hydrolysis of Fe(III) generated by the oxidation of Fe(II) [50].

Oxygen solubility presents a challenge in microbial cultures, particularly in species like *T. atroviride*, where the process is aerobic and oxygen transfer becomes a critical factor that requires special attention, as evidenced by the decrease in DO during the cultivation stage. This decrease is largely attributed to the specific morphology developed in liquid culture, such as pellet formation, which hinders efficient oxygen transfer [53]. However, excessive oxygen supply can be detrimental to the bioprocess, leading to damage or hindrance [54]. Moreover, the decrease and variation of DO in the medium are influenced by biomass production, resulting in a greater demand for air supply, and the synthesized metabolites can alter the viscosity of the culture medium, further affecting oxygen availability [55]. Wu et al. (2022) [56] suggested that the generation of micro-nanobubbles in a system could lead to oxygen enrichment in the water, potentially offering a solution to address the oxygen demand in microbial cultures. One of the main advantages of using such systems is that the mixing is induced by aeration, and the air used for mixing provides the necessary oxygen for the biological oxidation and growth of microorganisms [11].

The oxidation-reduction potential is one of the parameters that exerts a significant influence on the interaction between *Trichoderma* and metal ions [57], as it determines the conditions for accepting electrons (reduction) or donating electrons (oxidation) in a system [58]. In this study, both systems exhibited oxidizing conditions. However, they displayed differences, with the TR-MNBs system being significantly (*p* < 0.05) superior to the FS system. The oxidizing conditions in the TR-MNBs system can be attributed to the interaction of MNBs and fungal biomass during fungal cultivation (Figure 7). When the MNBs burst, the gas-liquid interface promotes the chemical reactions, leading to a drastic change and stimulation of the production of reactive oxygen species (ROS) such as hydrogen peroxide (H_2_O_2_), radical anion superoxide (^−^·O_2_), and hydroxyl radicals (·OH), which have high oxidizing power [59]. Hence, in oxidizing conditions, ·OH may facilitate changes in the oxidative states of metal ions [60,61].

In this case, As(III) and Fe(II) could undergo oxidation to form As(V), and Fe(III), ultimately leading to generation of bioscorodite. The substantial fungal growth was found to be closely associated with glucose consumption. These two parameters exhibited a negative correlation (*p* < 0.05 in TR-MNBs and *p* < 0.005 in SF), underscoring the efficiency of both systems in biomass production. The glucose consumption also underscores *T. atroviride*’s ability to thrive under challenging conditions, including the presence of elevated arsenic concentrations. Part of the tolerance observed in fungal species can be attributed to their capacity for diverse physical-chemical reactions, encompassing oxidation, reduction, methylation, alkylation, intracellular immobilization of metal ions, or physical precipitation [62].

Another mechanism used for ion immobilization is through biomineralization, which typically involves the release of ligands from cellular or extracellular substances that undergo precipitation through the redox process [63]. Moreover, specific concentrations of metal ions can induce changes in the metabolism and the growth and morphology of the microorganism [64].

*Trichoderma* has been recognized as an efficient fungus for mycoremediation due to its favorable characteristics, including a rapid growth rate, complex and extensive network of hyphae, production of extracellular enzymes, capability to interact with complex contaminants, adaptability to pH, and temperature fluctuations [57]. These properties have been taken into consideration by our research group for the disposal of arsenic through scorodite formation, which offers several advantages over other biological processes that predominantly involve thermoacidophilic microorganisms. Unlike those processes, scorodite formation does not necessitate for long-time high temperatures or chemical oxidants for the precipitation process [65].

Scorodite is considered one of the suitable forms for the final disposal of As, owing to its high thermodynamic stability [66]. As(III) is the predominant species in metallurgical residues, and the production of bioscorodite has been considered as a strategy for As removal [67]. Previous studies have evaluated the crystallization of scorodite using *Acidianus sulfidivorans* in the absence of primary minerals or seed crystals [12]. However, in conventional scorodite synthesis, the addition of seed crystals, such as scorodite or other heterogeneous crystals, has been observed to significantly promote scorodite formation. This is because seed feeding provides abundant active sites that facilitated the union and growth of this mineral on their surfaces [68,69]. Similarly, Tanaka and Okibe (2018) [70] have described a process for scorodite production through the oxidation of As and Fe using the extremophile archaea *Acidianus brierley*. Additionally, Vega-Hernández et al. (2020) [67] have reported the use of granular activated carbon to oxidize As under thermoacidophilic conditions in the presence of air.

The production of scorodite is a highly pH-dependent process, where pH influences the rate formation [46]. It is crucial to maintain stable pH during the reaction to prevent fluctuations, as the crystallization of ferric arsenate can generate H^+^ in the reaction system [47]. Fujita et al. (2009) [71] have noted that scorodite precipitation and arsenic leachates can form at a pH of 1.2. They also mentioned the possibility of jarosite (KFe_3_(SO_4_)_2_(OH)_6_) formations overlapping with scorodite formations under acidic conditions, although the formation of this mineral becomes unfavorable at a pH below 1. Min et al. (2005) [72] indicated that scorodite can be produced at a pH below 5, with optimal crystallization occurring within the pH range of 2–4. In addition, Kitamura et al. (2016) [48] reported fluctuations in the ORP during the scorodite producing process, reaching up to 280 mV. However, Guo and Demopoulos (2018) [73] highlighted that one essential condition for the mineral formation is the presence of a minimum oxygen concentration.

The characterization of chemical scorodite and bioscorodite using XRD, SEM, and FTIR techniques revealed similarities with scorodites previously documented by various researchers. Specifically, in the FTIR analysis, bands were identified between 510–420 cm^−1^, associated with the As-O bonding present in pure mineral scorodite [12,43].

However, during the FTIR analysis of the scorodite in the TR-MNBs system, a low absorbance was observed at 2900 cm^−1^, corresponding to the stretching vibrations of C-H [74]. Additionally, two other signals within the wavelength range of 3300–3600 cm^−1^ had characteristic peaks of römerite [43]. This change in absorbance could be attributed to the presence of oxygen and the incorporation of varying amounts of Fe. Furthermore, it is inferred that römerite is not the only mineral forming during the reaction, as there are other minerals, such as coquimbite (Fe_2_(SO_4_)_3_·9H_2_O), halotrichite (FeAl_2_(SO_4_)_4_·22H_2_O), and copiapite (MgFe[(OH)(SO_4_)_3_]_2_·218H_2_O) [75].

Another mineral that has the potential to form during the culture is ferrihydrite, especially at high iron concentrations (As:Fe molar ratio < 0.4) and elevated oxidation-reduction potential (ORP) [76]. These minerals might have contributed to arsenic removal through adsorption-precipitation processes. Reports by Vega-Hernández (2021) [44] and Ramirez-Castillo et al. (2023) [19] have discussed solid precipitation. In the SF culture, solid formation occurred and obtained 30% of arsenic removal at day 5. This removal could be attributed to an irreversible As adsorption, facilitated by a gradual build up of As(V) complexation on the surface of non-exchangeable sites of Fe(III) derivatives like ferrihydrite, followed by the addition of Fe(III) ions in solution, as reported by previous studies [77]. Ferrihydrite formation could have taken place during the culture, facilitated by the attained high ORP (244 mV), low pH (2.7), and molar As:Fe ratio of 0.14, as previously reported [76].

The low concentrations of Fe(III) and As(V) remaining after precipitation significantly hindered bioscorodite formation, as indicated by the small peaks in the XRD diffractogram (Figure 5a) and FTIR spectrum (Figure 6a). In contrast, an analysis of crystals from the fungal culture in the TR-MNBs system clearly showed bioscorodite formation, facilitated by the high (70%) As removal at the 5-day mark that was adsorbed to the ferrihydrite-MNBs-biomass complex. This removal was likely due to the rapid complexation of As(V) with the exchangeable sites of the ferrihydrite-MNBs-fungal biomass complex. The weak complex formation in the TR-MNBs culture likely allowed As and Fe availability for bioscorodite formation, unlike the complex and stable precipitate solids produced in the SF culture. Excess Fe(III) ions in both cultures possibly contributed to the formation of römerite and other minerals, alongside bioscorodite.

Recent studies have highlighted the catalytic properties of iron oxides in the oxidation and reduction of species such as Fe(II) and As(III), employing two proposed mechanisms. The first involves adsorption and direct heterogeneous oxidation via dissolved oxygen (DO), while the second operates through electrochemical catalysis [50]. In the case of this methodology involving oxygen supply through MNBs in the fungal culture, the MNBs could potentially generate ·OH radicals, facilitating oxidation-reduction reactions for As(V) and Fe(III) ion formation, as well as ferrihydrite complexation with As(V). This, in turn, aids stable bioscorodite formation. This mechanism might operate similarly or involve complex formation on the surface of reduced species [50,76,77].

## 5. Conclusions

Aerating the *T. atroviride* culture with MNBs facilitated the oxidation of As(III) and Fe(II), as well as the production of bioscorodite, römerite, and ferrihydrite minerals, even when confronted with high concentrations of ferrous iron, low pH, and elevated ORP (306 mV) within the fungal culture. Despite both systems displaying a decrease in DO, pH, and glucose concentration, which correlated (*p* < 0.05) with an increase in glucose oxidase activity, the MNBs showed a positive impact on the ORP, fungal biomass, and arsenic removal. This was followed by the rapid complexation of As with the surface of ferrihydrite, facilitating a substantial arsenic removal rate of 70% after 5 days. In contrast, the shaking flask culture exhibited a lower arsenic removal value of 30%, along with the observation of significant precipitated solids. This was attributed to the slow formation of As(V) bond-to-bond complexes with non-exchangeable ferrihydrite sites and Fe(III) ions.

Both bioscorodite samples obtained from the acidic and oxidizing fungal culture, when subjected to heating at 90 °C for 32 h, resulted in the generation of bioscorodite and römerite crystals. These were characterized through FTIR, SEM, and XRD analyses. Scorodite formation was attributed to *T. atroviride*’s high tolerance to thrive under stressful conditions, including elevated iron concentrations and the presence of the highly toxic arsenic ion As(III), since initial time of cultivation. In the case of the RT-MNBs system, aeration with MNBs further enhanced these abilities. For future research endeavors, it is recommended to optimize the conditions of the TR-MNBs aeration system and explore the application of this methodology in a pilot treatment plant with immobilized fungus.

## Figures and Tables

**Figure 1 jof-09-00857-f001:**
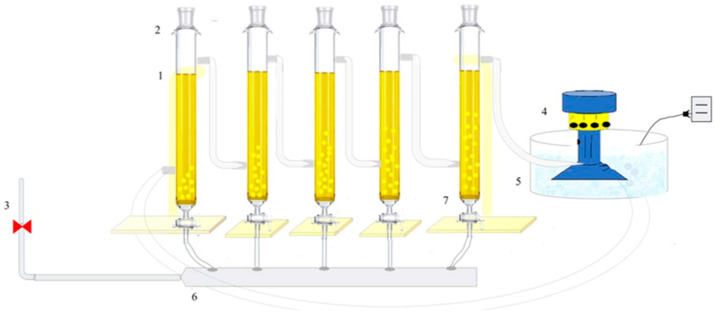
Tubular reactor with micro-nanobubbles generation. Components: Tubular glass columns (1), condensers (2), air supply (3), pump (4), temperature control system (5), hose (6), and ceramic diffuser of <0.1 µm of porous size (7).

**Figure 2 jof-09-00857-f002:**
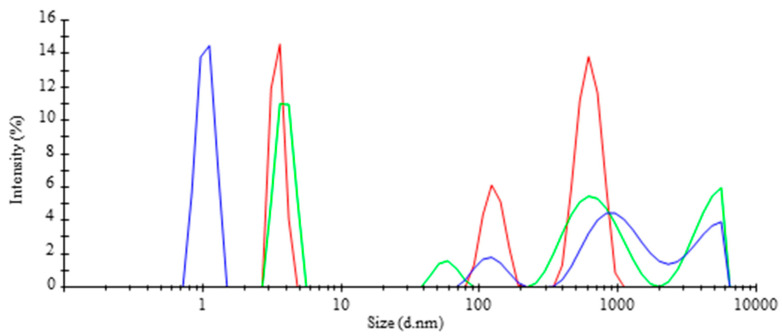
Intensity as a function of the size distribution of bubbles in the tubular reactor.

**Figure 3 jof-09-00857-f003:**
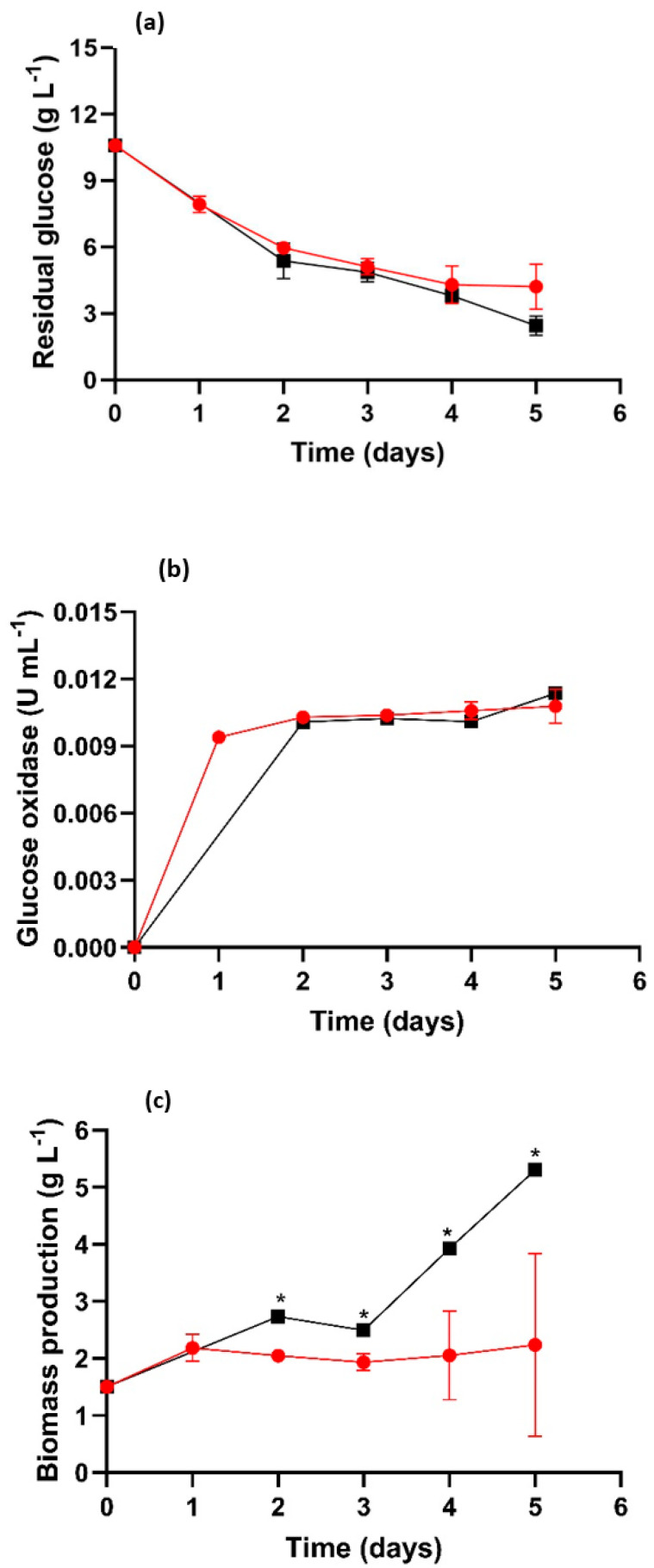
Residual glucose (**a**), glucose oxidase activity (**b**), and fungal biomass production (**c**). SF (
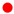
), TR-MNBs system (
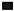
). * Indicates significant difference (*p* < 0.05).

**Figure 4 jof-09-00857-f004:**
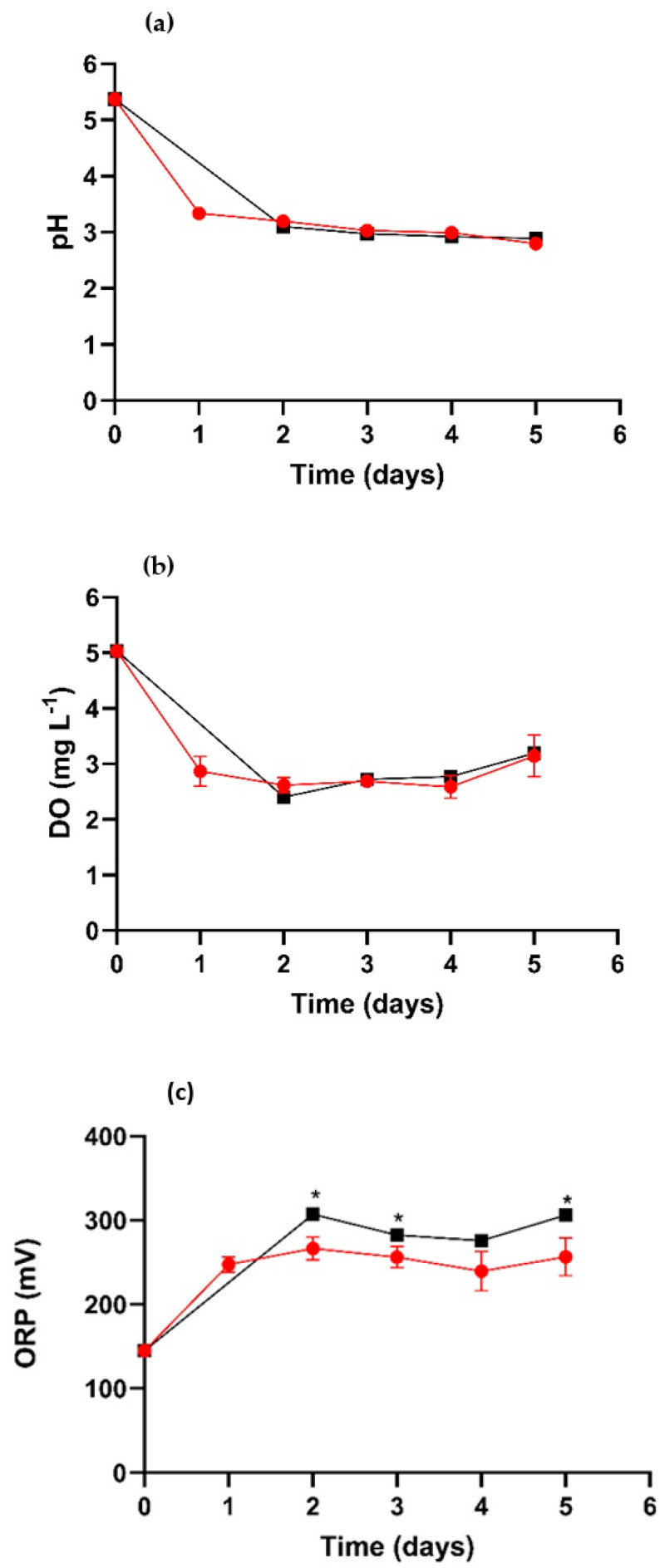
Physicochemical parameters during fungal growth: pH (**a**), DO (**b**), and ORP (**c**). SF (
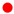
), TR-MNBs system (
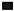
). * Indicates significant difference (*p* < 0.05).

**Figure 5 jof-09-00857-f005:**
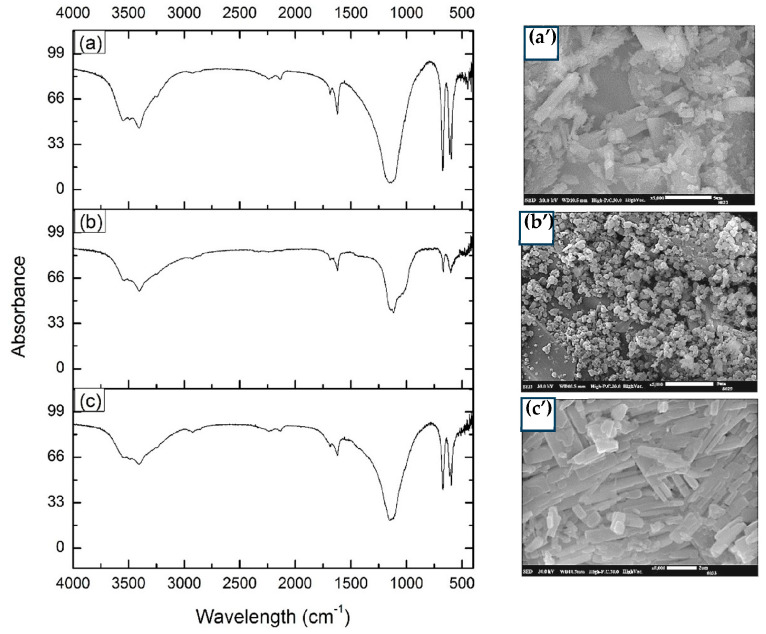
FTIR and SEM spectra of bioscorodite in SF at 120 rpm (**a**,**a**’), bioscorodite in TR-MNBs (**b**,**b**’), and scorodite chemically synthesized (**c**,**c**’).

**Figure 6 jof-09-00857-f006:**
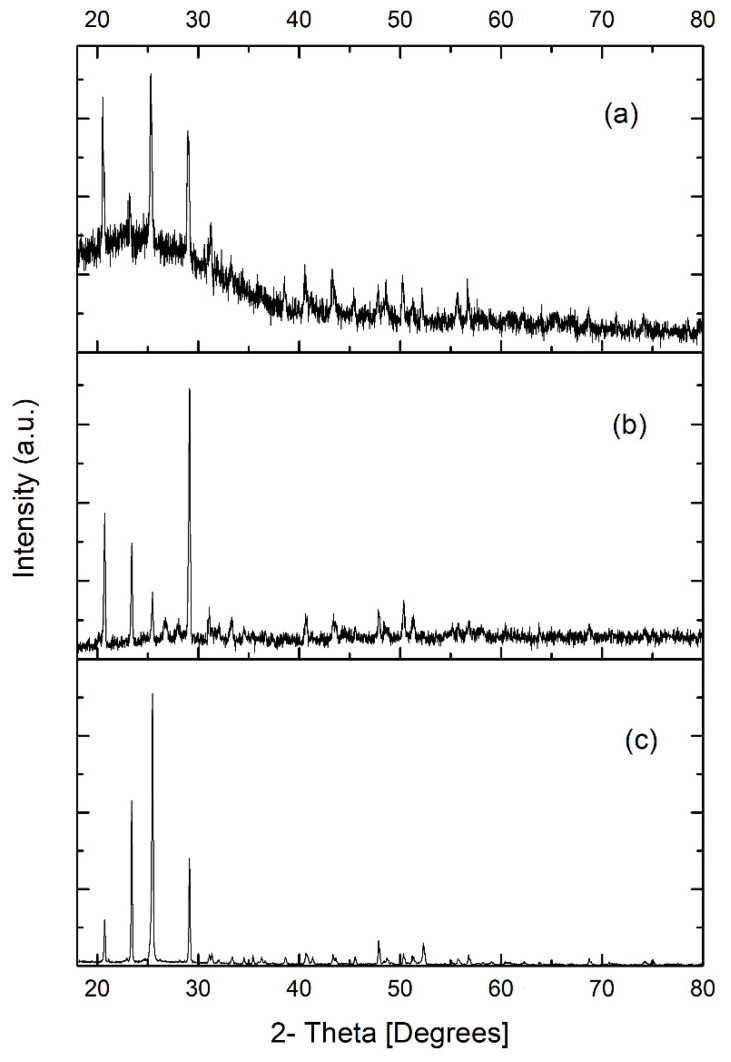
XRD diffractograms of scorodites are presented for the following conditions: bioscorodite from the SF system at 120 rpm (**a**); bioscorodite from the TR-MNBs system (**b**); and scorodite chemically synthesized (**c**).

**Figure 7 jof-09-00857-f007:**
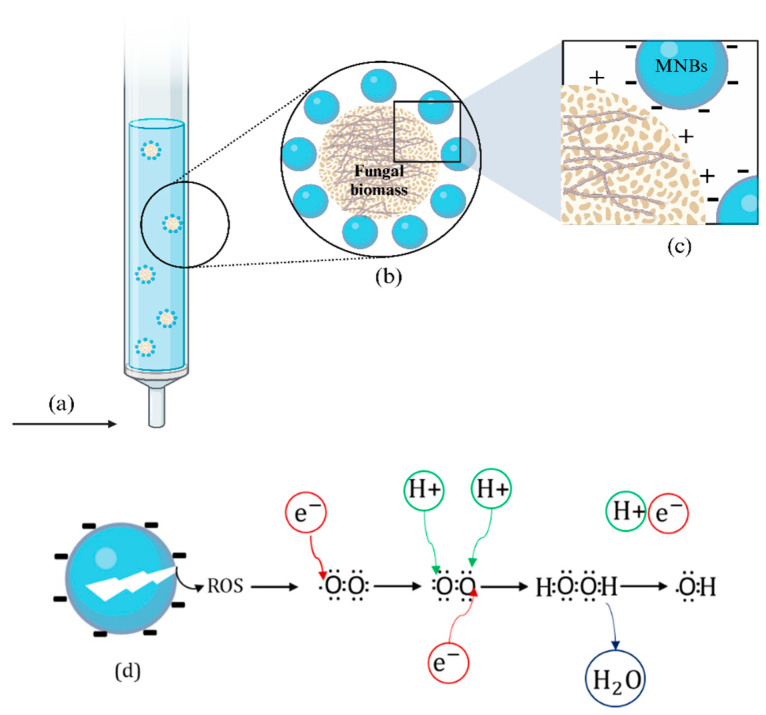
Interaction of micro-nanobubbles with fungal biomass: air supply (**a**); MNBs on the surface of fungal biomass (**b**), a close-up of the interaction between negatively charged MNBs (**c**) MNBs bust for reactive oxygen species (ROS) generation (**d**).

**Table 1 jof-09-00857-t001:** Average size diameter and intensity of the bubbles generated in the tubular reactor.

Average of Size Diameter (nm)	Intensity (%)	SD (±)
3.5	30.4	0.35
127.1	19.3	21.12
625.8	50.3	121.70

SD = Standard deviation.

## Data Availability

Not applicable.

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
