# Peer review of "Effect of Micro-Nanobubbles on Arsenic Removal by Trichoderma atroviride for Bioscorodite Generation"

_jof, 2023, doi:10.3390/jof9080857_

Round 1
Reviewer 1 Report
Dear Authors, thak you for your work. Results are supported by the data and the manuscript is well written. The work is of scientific interest and to me within the scope of the journal. Therefore i recommend your work to the editor for publication. With my best regards.
Author Response
Many thanks for your suggestions.
Reviewer 2 Report
Dear Authors, this study is very interesting and current as the presence of arsenic in drinking water is a real and widespread danger. Your manuscript indicated the effect of micro-nanobubble oxygenation on fungal culture parameters by generating biogenic scorodite (FeAsO4 ● 2H2O) using As3+ and Fe2+, indicating the possible formation of As-organic complexes. However, it is necessary to indicate if and how it is possible to extend these studies to the reduction of As in drinking water (scaling up). Best wishes.
Author Response
Many thanks for your kind comments
Reviewer 3 Report
This interesting study looks into the effect of micro-/nano-aeration on the culturing and myco-remedial potential of Trichoderma atroviride culture for bioscorodite generation. That said, this reviewer found the experimental approach difficult to follow. The likely culprit is the poor description of the methods and a poor overview of the experimental setup. Poor scientific English also contributed to a skewed interpretation of the work. Thus, I recommend completely revising the writing style and scientific English use to improve the manuscript's readability. And "Various culture conditions were evaluated using experimental designs in stirred flasks to enhance the oxidizing for the conversion of As (III) and Fe (II) into As (V) and Fe (III), as well as to lower the pH to acidic values." - lower pH means more acidic; what is 'oxidizing for the conversion of'? Again, a nonsense sentence.
Detailed comments and concerns include;
1. Re-write the paragraph from lines 88 to 102 to ensure meaning and to make the aims/objectives of the work as clear as possible. Sentences, and I quote, "Recently, our research group, Ramírez-Castillo et al. (2023) [19] employed an alternative approach by culturing the ascomycete fungus Trichoderma atroviride." are without logic and meaning.
2. The paragraph on micro-nano bubble formation (lines 103-112) seems out of place and might be better positioned at the start of the Discussion. It is more important to elaborate on the Aim of the work (the hypothesis) and to present the objectives (how the hypothesis is to be addressed). The present objective is "to determine the effect of oxygenation with micro-nanobubbles in bubble column reactors on the oxidative conditions and acidity using the fungus T. atroviride for the formation of bioscorodite." which seems to lack focus and does not accurately address the experiments as presented. Revision of the objectives is critical.
3 . Where exactly is the Center for Research and Advanced Studies of the National Polytechnic Institute (CINVESTAV) located? Give the full postal address.
4. Section 2.2 is titled "Characterization of Oxygenated Bubble Columns with Micronanobubbles" but this section instead describes the set-up of the bubble columns. And the description and image need to be completed. A much better idea and descriptive text are required in order to aid the reader in understanding this experimental setup. For example, the Section title indicates that it is air supply, but the title is oxygenation. Where does the nitrogen go? In the introduction, the reader is informed of all the ROS produced. What of these, and was there any effort to measure ROS in the present work?
5. In Section 2.3, what is meant by "the fungal culture were evaluated using two systems: SF and TRMNB with modified Wunder medium ..."? What is SF and TR-MNB? Please define each fully and/or explain the difference between the two. And the sentence "Particularly, in the SF system, 100 mL of culture medium was added to 250 mL Erlenmeyer flasks and 0.15 g of biomass on a dry basis, previously filtered." - what does this mean? What is its purpose? To say what? And the phrases " Samples were collected every 24 h from each replicate. On the other hand, 200 mL of the above-mentioned medium was added to the bubbling columns and 0.3 g of biomass was added. The culture was monitored every 24 hours from day 2 until 5 days." seem confused or do not address the procedure.
6. In the Results, what is the purpose of Figure 3? What do the red, green and blue lines represent in this graph?
I have provided sufficient examples of where this manuscript fails. Not so much in the science, although this is difficult to determine in the absence of the detail, but in the presentation. The latter leaves the reader astonished at the very weak elaboration of the work supposedly carried out—generally, a poor effort.
Reconsider after major revision!
English and general scientific copy editing is an absolute requirement. Many sentences are incomplete, lacking a verb/noun and in some places meaningless. Much stricter reading is needed - both by the authors and the journal.
Author Response
Thank you for your interesting suggestions to enhance the quality of the manuscript.
The main changes in the manuscript were made to the title, objective, and methodology, particularly in the microbiological section. Additionally, improvements were made to the results to enhance the understanding of the investigation on scorodite production in water contaminated with As(III) and Fe(II) aerated with micro-nanobubbles.
This research represents a novel contribution for dealing with this type of toxic metal, as it demonstrates the sustainable removal of As(III) using fungi. The use of fungi led to changes in the physicochemical properties of the culture medium, enabling in situ oxidation of As(III) and Fe(II) and their subsequent removal through biological mineral production. This alternative process offers a potential solution for the disposal of As in contaminated water.
will be send again for English editing work if it is required.
Attached to this are our answers

Round 2
Reviewer 3 Report
This revision is an improvement. There remain several issues in the presentation, though. First, the Figure numbering is incorrect. The caption to Figure 1 needs a more complete and informative description. What is the reader looking at?The caption to Figure 2 should read Size distribution and intensity of bubbles in the tubular reactor. Figure heading for Fig 8 (given as Fig 6) is obscure and needs elaboration. The alignment of graphs in Figures 3. 4 & 6 must be adjusted so that ALL panels align accurately. Is Fig 7 (currently Fig 8) necessary? Perhaps not - so consider removing it. The caption to Fig 8 (currently Fig 6) is inappropriate and lacks the detail required. Please explain what this Figure shows in the caption.
In the title, why is it important to state "aeration"? Was it the air or the micro-nano bubbles that elicited the effects?
In the Abstract, why is the aim given in the opening sentence? Typically Abstracts are written according to the scientific method and must include the following; Background, Methods used, Results and Conclusions and in that order. Please re-write accordingly.
In the Results, address the above concerns with Figure captions, numbering, and Table captions. Fig 3 the order seems incorrect. First, glucose is used by the enzyme glucose oxidase to drive growth and biomass accumulation. The correct order, please. For Fig 4, change the caption to read 'Physicochemical parameters during fungal growth'. In the captions to Figures 5 & ^, change biogenic to biogenerated. Fig 8 (or 7) can be omitted. A complete description is required for the caption to Fig 6 (8). Other recommended changes are indicated in the attached file.

Nil
Author Response
Answers to the questions given by the reviewer 3.
Querry: First, the Figure numbering is incorrect.
Answer: THE FIGURE NUMBERING HAS BEEN CORRECTED.
Querry: The caption to Figure 1 needs a more complete and informative description. What is the reader looking at?
Answer: THE CAPTION FOR FIGURE 1 HAS BEEN REVISED WITH A MORE DETAILED DESCRIPTION.PLEASE REFER TO THE MANUSCRIPT, HIGHLIGHTED IN RED (Lines 131-133).
Querry: The caption to Figure 2 should read Size distribution and intensity of bubbles in the tubular reactor.
Answer: THANK YOU FOR YOUR COMMENT. THE CAPTION FOR FIGURE 2 HAS BEEN UPDATEED TO "INTENSITY AS A FUNCTION OF THE SIZE DISTRIBUTION OF BUBBLES IN THE TUBULAR REACTOR." (Line 244 o Figure 2, and line 245 in Table 1).
Querry: Figure heading for Fig 8 (given as Fig 6) is obscure and needs elaboration.
Answer: APPRECIATE YOUR INPUT. WE HAVE ENHANCED THE QUALITY OF FIGURE 8 AND REVISED ITS CAPTION FOR CLARIFY. (Lines 413-415). Now figure 5. The figure caption has also been amended and highlighted in red..
Querry: The alignment of graphs in Figures 3. 4 & 6 must be adjusted so that ALL panels align accurately.
Answer: THE GRAPHS IN FIGURE 3, 4, AND 6 HAVE BEEN ADJUSTED TO ENSURE ACCURATE ALIGNMENT.
Querry: Is Fig 7 (currently Fig 8) necessary? Perhaps not - so consider removing it.
Answer: THANK YOU FOR YOUR SUGGESTION. FIGURE 8 HAS BEEN REMOVED, AND THE RELEVANT RESULTS ARE NOW DISCUSSED IN THE TEXT (Lines 445- 448).
Querry: The caption to Fig 8 (currently Fig 6) is inappropriate and lacks the detail required. Please explain what this Figure shows in the caption.
Answer: THE FEEDBACK IS VALUED. FIGURE 6 HAS BEEN MODIFIED (Lines 496-509) AND RENAMED, AS FIGURE 7. (Lines 511-514). “Interaction of micro-nanobubbles with fungal biomass: air supply (a); MNBs on the surface of fungal biomass (b), a close-up of the interaction between negatively charged MNBs and a portion of positively charged biomass, followed by MNBs burst leading to the production of reactive species (ROS).
Querry: In the title, why is it important to state "aeration"? Was it the air or the micro-nano bubbles that elicited the effects?
Answer: TO ADDRESS YOUR CONCERN, THE TITLE HAS BEEN REVISED TO AVOID CONFUSION: “Effect of micro-nanobubbles on arsenic removal by Trichoderma atroviride for bioscorodite generation".
Querry: In the Abstract, why is the aim given in the opening sentence? Typically Abstracts are written according to the scientific method and must include the following; Background, Methods used, Results and Conclusions and in that order. Please re-write accordingly.
Answer: THANK YOU FOR YOUR SUGGESTION. THE ABSTRACT HAS BEEN RESTRUCTURED IN ACCORDANCE WITH THE SCIENTIFIC METHOD, PRESENTING BACKGROUND, METHODS, RESULTS, AND CONCLUSIONS (Lines 17-36).
Querry: In the Results, address the above concerns with Figure captions, numbering, and Table captions. Fig 3 the order seems incorrect. First, glucose is used by the enzyme glucose oxidase to drive growth and biomass accumulation.
Answer: YOUR INPUT HAS BEEN INCORPORATED. THE RESULTS SECTION NOW ADDRESSES THESE CONCERNS BY REORDERING AND PROVIDING CLEAR CAPTIONS FOR FIGURES AND TABLES, GLUCOSE OXIDASE ENZYME ACTIVITY, GLUCOSE AND BIOMASS (Lines 157-174).
Querry: The correct order, please. For Fig 4, change the caption to read 'Physicochemical parameters during fungal growth'. In the captions to Figures 5 & ^, change biogenic to biogenerated. Fig 8 (or 7) can be omitted. A complete description is required for the caption to Fig 6 (8). Other recommended changes are indicated in the attached file.
Answers: THANK YOU FOR YOUR GUIDANCE. THE CAPTION FOR FIGURE 4 HAS BEEN ALTERED. "Physicochemical Parameters During Fungal Growth" (Lines 363-364). The terms "biogenic" have been updated to "biogenerated" in Figures 5 and 7. Figure 8 has been omitted, and the caption for Figure 6 (now Figure 7) has been thoroughly described (Lines 414-415) and figure 7 (Lines 496-509).
FIGURE 8, ACTUAL NUMBER IS, Figure 7. Interaction of micro-nanobubbles with fungal biomass: air supply (a); MNBs on the surface of fungal biomass (b), a close-up of the interaction between negatively charged MNBs and a portion of positively charged biomass, followed by MNBs burst leading to the production of reactive species (ROS).
IN ADDITION, TO AVOID CONFUSION, THE WORDS BIOGENIC, AND BIO-SCORODITE WERE CHANGED IN THE MANUSCRIPT BY BIOSCORODITE.
